# Significance of Artificial Intelligence in the Study of Virus–Host Cell Interactions

**DOI:** 10.3390/biom14080911

**Published:** 2024-07-26

**Authors:** James Elste, Akash Saini, Rafael Mejia-Alvarez, Armando Mejía, Cesar Millán-Pacheco, Michelle Swanson-Mungerson, Vaibhav Tiwari

**Affiliations:** 1Department of Microbiology & Immunology, College of Graduate Studies, Midwestern University, Downers Grove, IL 60515, USA; jelste@midwestern.edu (J.E.); mswans@midwestern.edu (M.S.-M.); 2Hinsdale Central High School, 5500 S Grant St, Hinsdale, IL 60521, USA; akash.r.saini@gmail.com; 3Department of Physiology, College of Graduate Studies, Midwestern University, Downers Grove, IL 60515, USA; rmejia@midwestern.edu; 4Departamento de Biotechnology, Universidad Autónoma Metropolitana-Iztapalapa, Ciudad de Mexico 09340, Mexico; ama@xanum.uam.mx; 5Facultad de Farmacia, Universidad Autónoma del Estado de Morelos, Av. Universidad No. 1001, Col Chamilpa, Cuernavaca 62209, Mexico; cmp@uaem.mx

**Keywords:** virus entry, virus–host cell interactions, entry receptors, artificial intelligence

## Abstract

A highly critical event in a virus’s life cycle is successfully entering a given host. This process begins when a viral glycoprotein interacts with a target cell receptor, which provides the molecular basis for target virus–host cell interactions for novel drug discovery. Over the years, extensive research has been carried out in the field of virus–host cell interaction, generating a massive number of genetic and molecular data sources. These datasets are an asset for predicting virus–host interactions at the molecular level using machine learning (ML), a subset of artificial intelligence (AI). In this direction, ML tools are now being applied to recognize patterns in these massive datasets to predict critical interactions between virus and host cells at the protein–protein and protein–sugar levels, as well as to perform transcriptional and translational analysis. On the other end, deep learning (DL) algorithms—a subfield of ML—can extract high-level features from very large datasets to recognize the hidden patterns within genomic sequences and images to develop models for rapid drug discovery predictions that address pathogenic viruses displaying heightened affinity for receptor docking and enhanced cell entry. ML and DL are pivotal forces, driving innovation with their ability to perform analysis of enormous datasets in a highly efficient, cost-effective, accurate, and high-throughput manner. This review focuses on the complexity of virus–host cell interactions at the molecular level in light of the current advances of ML and AI in viral pathogenesis to improve new treatments and prevention strategies.

## 1. Diverse Landscape of Virus–Host Cell Interactions

Viral entry into a host cell is a complex and highly dynamic stepwise process initiated when viruses engage with the host cell receptor in a susceptible target cell [1]. However, there is no viral entry if the virus encounters a cell lacking a receptor. Hence, the presence of host cell receptors is a critical determinant of virus entry and subsequent processes related to virus replication [2,3,4]. This basic notion that viruses utilize specific cell receptors for entry has been demonstrated using resistant cell lines that lack entry receptors and through genetic knockout studies [5,6,7]. The observation that transient expression of the target receptor in a given resistant cell line makes it susceptible to that virus further supports this notion. For example, Chinese hamster ovary cells (CHO-K1) lack entry receptors for herpes simplex virus type 1 (HSV-1); therefore, they are resistant to HSV-1 infection. The screening of cDNA libraries from CHO-K1 cells using high-throughput technologies led to the identification of multiple cell receptors for HSV-1 entry [8,9,10]. For these reasons, CHO-K1 cells constituted the ideal experimental model for identifying cell receptors for different viruses, such as Ebola virus and coronaviruses [11]. The other advantages of using CHO-K1 cells lay in their rapid growth (either as adherent cells or in suspension); their high DNA transfection efficiency; and the ability to genetically manipulate proteins, antibodies, and growth hormones [12,13]. 

Traditional (wild-type, clinical isolates) and nontraditional (pseudotyped) viruses are used to study virus–host tropism and functional receptor interactions in a given research setting. The viruses have evolved multiple strategies to enter target cells by first attaching or binding to the targeted host cell. This process can bring multiple virions to attach and bind to achieve viral entry [14,15]. This is the case for negatively charged heparan sulfate proteoglycans (HSPGs), which are ubiquitously located on the cell surface and embedded in the extracellular matrix. A negatively charged HSPG provides docking sites for positively charged viral glycoproteins via ionic interactions [16]. Interestingly, in the case of severe acute respiratory syndrome coronavirus 2 (SARS-CoV-2), it has been shown that binding of HSPG to the receptor binding domains (RBDs) on the spike glycoproteins is essential for SARS-CoV-2 binding and entry, particularly in cells that express low levels of ACE2 [17,18,19]. In addition, besides enhancing viral attachment to cells, HSPGs also regulate the conformational change in the spike glycoprotein to favor viral entry. Multiple studies have also demonstrated that the HSPG also stabilizes the RBD in the exposed “open state” conformation to interact with the ACE2 receptor with higher affinity to promote viral entry [20,21,22,23,24]. The above findings open up a new avenue to target HSPGs by conjugating HS-targeting compounds to a spike-neutralizing antibody in therapeutic development with anti-HS-based molecules [25]. However, caution would be required to test whether such therapy brings unwanted side effects to the host cell, since HSPGs are endogenously expressed in multiple cell types as housekeeping genes [26]. 

Given the molecular diversity of glycans and their significance in host–pathogen interactions, a recent study by Bojar et al., 2021 attempted to use natural language processing (NLP) to develop deep-learning models for glycans to predict glycan immunogenicity, pathogenicity, and glycan-mediated immune evasion via molecular mimicry to understand the host-microbial relationship [27]. In addition, the language model SweetTalk was utilized by one study for the analysis of information-rich glycosaminoglycans to predict, for instance, viral binding to host cell [28]. Using pattern-learning algorithms, such as machine learning (ML), to identify key glycan motifs involved in virus–host cell interactions would generate significant interest in the field to address virus-tropism, evolutionary changes in glycan phenotypes for predictive purposes in the context of host-microbe interactions, diagnostic fragment identification, and high-throughput glycomics [27,29].

Multiple lines of evidence indicate that cell surface heparan sulfate (HS) on actin-rich filopodia also provides a surfing mechanism for the viruses to reach the base of the target cells where receptors are centered and concentrated [30,31,32]. Using live cell imaging strategies and the availability of fluorescently tagged viruses, we and others have analyzed the movement of viruses from the tip to the base of filopodia [33,34,35,36]. In addition, multiple viruses show a redundancy in HS usage, highlighting the advantage of overtaking the cellular machinery using an archaic and evolutionary conserved molecule [37]. HS receptor usage is also known to be a key determinant of cross-species transmission and human disease potential [38,39,40]. Therefore, it is crucial to have a detailed understanding of HS or other involved glycan-mediated mechanisms, which dictate cross-species transmission of newly emerging zoonotic diseases. This is true not only for viral surveillance but also for evaluating potential new virus emergence. The shared expression of HS and sulfated forms of HS among multiple species may allow new virulent strains to evolve into another zoonotic transmission for both natural and domesticated animals [37,41,42,43]. Interestingly, a recent ML study built a dataset that predicted that enveloped viruses tend to display a broader repertoire of receptor usage than non-enveloped viruses [44]. This study used a gradient boosting model (GBM) to identify plasma membrane-associated proteins located at the cell surface (surfaceome) that are more likely to be used as virus receptors. Since envelope proteins are structurally less rigid and more flexible than capsid proteins, this feature may allow envelope proteins to evolve and add genetic diversity to facilitate using a new receptor. Using alternative receptors sufficient for viral entry could benefit enveloped viruses to infect more host species and favor cross-species transmission [44]. After the viral binding to the host cell, virus glycoproteins selectively recognize host cell protein receptors and sulfated moieties in the HS chain via conformational shifts, mediating virus–host cell fusion. The latter process happens at the host cell surface and is not affected by low pH; hence, it is called pH-independent or non-endosomal route of viral entry. However, if virus–cell fusion is mediated by low pH in an endosomal vesicle, it is termed a pH-dependent or endosomal mode of viral entry [45]. In this process, the viruses are transported either in a clathrin-coated pit or via micropinocytosis or caveolae, with Rho GTPases playing a significant role [46,47]. A detailed understanding of these mechanisms will generate huge clinical and epidemiological benefits. For instance, the viral entry process constitutes an attractive target for disease prevention because the virus remains in the extracellular matrix during this phase and has not yet utilized any host cellular machinery for gene expression or genome replication. Consequently, it is easier to design and screen drug molecules targeting the virus at this extracellular stage than when it has already penetrated the cell. Since many viruses and bacterial pathogens widely utilize HS chains, it represents one of the best strategies for designing a broad-spectrum antimicrobial pharmacological intervention. On the other end, it is clear that based on the localized surface expression, the selective affinity of virus envelope glycoprotein for a receptor, efficiency, and the mode of entry cannot be generalized for all the viruses as it varies being a multifactorial event (Figure 1).

Despite the complexity of viral cell entry, the process remains essential for understanding viral tropism, pathogenesis, cross-species transmissibility, and the design of novel antiviral drugs [48]. Competitive inhibition of viral entry with a molecule that binds ubiquitously to HS expressed on the cell surface can interfere with viral entry for many viruses, allowing the molecule to act as a broad-spectrum inhibitor. In the context of sexually transmitted infections (STIs), this strategy could be far more useful since many viruses and the bacterial pathogens involved in these conditions use conserved HS moieties to promote infection. Similarly, interfering with HS-virus interaction for viruses using HS as a co-receptor alone can have a significant impact. However, the enormous structural diversity of heparan sulfate and the precise glycan moieties involved in virus interactions in the context of medically important viruses have not yet been comprehensively mapped in detail. Therefore, this could restrict the targeted approach for some viruses. In the era of a pandemic, it is clear that receptor usage remains a key determinant of cross-species transmission and human disease potential. The attachment and entry of SARS-CoV-2 into the host cell is an outcome of such multivalent interactions between viral spike glycoprotein and cell membrane ACE2 receptor. Interfering with these recognition events, thereby blocking viral entry into host cells, is one of the most promising strategies for developing novel antiviral drugs to prevent SARS-CoV-2. Another aspect of SARS-CoV-2 biology to consider is the constant evolution in the diversity of the spike glycoprotein sequence, which displays an elegant mechanism on how to improve host adaptability, infection, transmissibility, and escape of immune response in variants of concern (VOCs) (Figure 2). Figure 2 shows the distribution of mutations in the genome of SARS-CoV-2 variants. It is clear that most of the mutations are found in the gene encoding the spike protein. This means that the spike protein gene is critically important in the prevalence of the virus because its infectivity depends on it. It should be noted that within this gene, the region with the highest incidence of mutations is the one encoding for the S1 subunit, responsible for interaction with and binding to the ACE2 receptor. It is also noteworthy that the amino acids Gly502, Tyr449, and Asn487 are the least mutated among the residues of the RBD domain of the SARS-CoV-2 spike protein, which are known to interact with the ACE-2 receptor and represent potential sites for the design of drugs.

Although the SARS-CoV-2 VOCs have evolved extensively due to numerous mutations of the spike glycoprotein, one variant carrying a single amino acid change (D614G, N501Y, E484K including delta P681R variant) has become the most prevalent form in the global pandemic with high viral infectivity [49,50] (Figure 2 and Figure 3). Such large transmissibility is due to mutations in the spike glycoprotein favoring interaction with host cell receptors with a higher affinity, which results in easier cell entry and a higher likelihood of escaping neutralizing antibodies [49,50]. Still, whether these spike variants exhibit a higher affinity for unidentified new receptors on the host cell surface is unknown. Consequently, a detailed understanding of the evolving key molecular motifs in the spike glycoprotein responsible for cross-species transmission of SARS-CoV-2 will be critically important. Such understanding will improve strategies for viral surveillance and facilitate the evaluation of potential new virus emergence [51]. Interestingly, a recent cross-sectional study explored a haplotype-based artificial intelligence model using more than 5 million viral sequences to identify emerging novel SARS-CoV-2 variants due to the acquisition of new mutations or a mixture of mutations from multiple variants [52]. Therefore, earlier detection of emerging novel SARS-CoV-2 variants is vital for public health surveillance of potential viral threats and for prevention strategies [52]. In addition, a deep learning model based on the graph convolutional network (GCN) found that exogenous substances (poor air pollution quality) affected the transcriptional expression of the ACE2 gene. These results were further confirmed by using quantitative polymerase chain reaction (qPCR) experiments, providing additional supporting evidence for indoor air pollutants identified by the GCN model [53]. Clinically, it has been shown that a higher ACE2 receptor expression is commonly seen in patients with smoking and chronic obstructive pulmonary disease [54]. In addition, the reported overexpression of Niemann Pick C1-like 1 (NPC1-L1) protein among obese population may also predispose to enhanced risk for SARS-CoV-2 infectivity since adipose tissue also has a higher expression of ACE2 [55,56]. Based on this, we hypothesize that the astonishing success of SARS-CoV-2 entry into human gastrointestinal cells and tissue with high ACE2 expression may also be compounded by the contributions of NPC1-L1 [57,58]. 

The periodic emergence of new respiratory viruses capable of spreading globally, threatening the human population and economy, is becoming real. A well-known example in this setting is influenza A viruses (IAVs), which use specific glycan receptors on airway epithelial cells to initiate entry and replication. As per the current model, avian viruses preferentially recognize α2,3-linked sialic acids abundant in the gastrointestinal tract of birds, while human IAVs have an affinity for α2,6-sialosides expressed on lung epithelial cells in our upper airways. A switch in HA specificity from α2,3- to α2,6-linked sialic acids is associated with increased infection and transmission in humans [59,60]. Therefore, usage and characterization of the glycan receptor-binding phenotype of IAVs could provide an early indicator of increased infection. In this direction, a study used support vector machine (SVM) learning, which enables the sialic acid receptor pattern recognition by IAVs [61]. On the virus end, hemagglutinin (HA) is a key viral glycoprotein responsible for facilitating IAV’s entry and infection by promoting the fusion between the host membrane and the virus [62]. In this direction, a recent study used ML methods such as K-nearest neighbor (KNN), logistic regression (LR), random forest (RF), and SVM to build the fundamental classifier model for the HA dataset to facilitate the identification and prediction of HA [63]. Although further validation of such modeling needs to be tested using vaccine design and or antiviral drug targets [63]. Taken together, the molecular mechanism of viral entry represents an exciting field of research to understand receptor expression and identify new and or alternative receptors that contribute to viral entry, host cell invasion, virulence, and tissue tropism, with the ultimate goal of developing strategies to block virus-receptor-specific events to prevent disease outcomes. 

To enter the host cell, viruses use multiple strategies [1,15,64]. The process brings the virus into close contact with the host cell receptor. The receptor recognition can vary depending on the type of host cell, viral tropism, and the presence of a co-receptor. When the interactions between viral attachment glycoproteins and their cognate receptors involve a small area of interaction between the viral and cellular receptors, it is termed a low-affinity (μM–mM) binding [65]. This process does not lead to conformational changes in the entry proteins. In contrast, high–affinity (nM–pM) interactions with virus and cell receptors involve a large area of interaction between the viral and cellular receptors and often involve significant conformational changes [15,46]. Although the initial low-affinity virus–cell interactions require mostly carbohydrates, heparan sulfated modified by 3-O sulfotransferase (3-OSTs) generates a glycoprotein D (gD) receptor for HSV-1 entry, which exhibits a similar high affinity for the receptor as a protein receptor [10,16,65]. The viral entry receptors range from cell adhesion molecules, immunoglobulin superfamily receptors, integrins, growth factors, and tumor necrosis factor receptor superfamily to the diverse array of heparan sulfate proteoglycans [16]. A virus–cell fusion begins after one or multiple viral glycoproteins engage with the functional host cell receptor displaying fusogenic properties [66,67]. This event involving virus–cell fusion is also a dynamic process triggering the exposure of an intermediate configuration state in the pre- and post-fusion forms of the fusion peptide in the virus glycoprotein, which engages the lipid bilayer and the associated trafficking proteins on the host cell [15,47]. This virus–cell fusion process offers a unique opportunity for preventing/therapeutic interventions by developing novel pharmacological agents or neutralizing antibodies that may serve as a key immunogen factor [2,68]. Interestingly, at some point during the process of virus attachment to the host cell, a co-receptor molecule may play a role in completing the virus entry process [1]. For example, in the case of human immunodeficiency virus (HIV) entry, the virus glycoprotein (gp) 120 first interacts with the CD4 receptor, which is expressed on helper T-lymphocytes, macrophages, and dendritic cells [69,70]. The initial high-affinity interactions between gp120 and CD4 bring conformational changes in the viral proteins that results in a second interaction of gp120 with chemokines (CXCR5 and CXCR4) receptors promoting virus–cell fusion [1,71,72]. Unlike HIV, the influenza virus glycoprotein hemagglutinin and the neuraminidase primarily use host cell receptor sialic acid to fuse to the host cell, avoiding the need for a co-receptor for cell entry [73,74]. On the other hand, the HSV-1 virus may independently utilize multiple cell receptors to gain cell entry. These receptors are not related to each other and include tumor necrosis factor (TNF) receptor family members, herpesvirus entry mediator protein (HVEM), immunoglobulin superfamily members nectin-1 and nectin-2, and the modified isoform of heparan sulfate proteoglycan 3-O-sulfated heparan sulfate (3-*O*S HS) generated by 3-O sulfotransferase [6]. In a unique case, it was recently shown that the receptor binding domain of spike protein of severe acute respiratory syndrome coronavirus 2 (SARS-CoV-2) specifically interacts with both the angiotensin-converting enzyme 2 (ACE2) receptor as well as a co-receptor HS. This dual binding results in the enhancement of viral entry (Figure 4) [17]. Despite the vast diversity of virus–host cell interactions, all virus and host cell proteins carry specific motifs or domains that perfectly enable them to interact in a lock–key model. 

## 2. Signaling Pathways That Promote Viral Entry into the Host Cell

The purpose of a virus having a close interaction with the host cell receptor is to hijack cells to maximize survival opportunities, preferably through a long-lasting and stable infection. A variant of this fate occurs when the virus induces an oncogenic response resulting in cancer of the infected tissue/organ (e.g., uterine cervix, liver, etc.). The perfect viral adaptation to fulfill its infective and replication goal is exemplified by various sophisticated signaling cascade strategies adapted by the human papillomavirus (HPV) and hepatitis B virus (HepB) [75,76]. In general, cellular signaling is activated upon viral interactions with the host cell receptor to induce receptor-mediated virus internalization. The very first virus-receptor interaction occurs either at the flat or ruffling membrane domains to anchor the virus and facilitate virus surfing and internalization [77,78,79]. In this direction, the creation of lab-based reporter viruses in a clinically relevant strain and their ability to preferentially infect target cells provide the necessary tools to study multiple steps involved in viral pathogenesis. For instance, using a lacZ (β-galactosidase) encoded reporter virus, the entry of the virus into the receptor-positive susceptible cells can be quantified (Figure 5A). The latter tool is also very useful, especially with high-throughput drug screening, to isolate and characterize compounds exhibiting promising effects in inhibiting viral infections [80,81]. Similarly, fluorescence-based reporter viruses can be used in parallel to assist in studying virus trafficking and localization, including the presence of unique virus surfing mechanisms on active filopodia in the preferentially targeted cell [31,78,79,82,83]. A representative Figure 5B (panels c and d) shows that HSV exploits actin-rich filopodia to surf and potentially spread quickly across cells. A similar process is utilized by many other viruses, including HPV and HepB [78,79]. The presence of fluorescence-tagged viruses also provides an excellent platform to understand the kinetics of virus pathogenesis in real time [83].

Our previous studies have shown that HSV-1 infection of the primary cultures of corneal fibroblasts results in the activation of PI3 kinase and Rho-GTPases cell signaling as soon as 10 min of exposure to the virus [82]. This observation was further supported by confocal and high-resolution electron microscopy imaging, which revealed changes induced by the viral exposure in the actin components of the cytoskeleton, such as stress fibers and filopodial extensions. The virus-activated signaling triggered by viral entry leads to modifications in the gene expression pattern, resulting in virus replication. Side effects of this viral control of intracellular signaling include activation of innate and adaptive immunity, growth, proliferation, survival, and apoptosis. Viruses also encode microRNAs (miRNAs) that regulate both the host and viral genes to generate a suitable environment for the virus by inhibiting host cell immune responses [85]. For instance, the hallmark of all herpesviruses is establishing lifelong latency in the host cell. In this regard, many herpesviruses encode viral microRNAs (vmiRNAs). Essentially, vmiRNAs enhance virus replication by avoiding the host defense system via regulating viral mRNAs and suppressing cellular mRNAs. Depending on the cell type and the kind of receptor utilized during cell entry, these signaling pathways may vary; however, the nature and function of cell signaling is to promote virus entry and gene expression for sustained viral control of the host cell. Viruses have also developed several strategies to exploit cell surface receptors and their signaling pathways to activate cellular events. These pathways include but are not limited to fibroblast growth factors (FGF), epidermal growth factor receptor (EGFR-tyrosine kinase), cellular integrins, cell adhesions molecules, G protein-coupled receptor (GPCR), and a variety of chemokines [86,87]. Since many viruses use GPCR mechanisms, these pathways constitute an attractive target for developing antiviral therapies [88]. Interestingly, it has recently been shown that SARS-CoV-2 uses toll-like receptor-4 (TLR-4) signaling pathway to enhance ACE-2 receptor expression, which increases virus entry [89]. Multiple DNA and RNA viruses use similar TLR signaling to invade the target cell [90]. In this direction, future studies focused on TLR pathways would improve our understanding of how to interfere with virus entry, trafficking, and reproduction. The final result could include vaccine formulation against TLR-conjugated virus-specific epitopes [90]. Taken together, the evidence makes it clear that multiple aspects of viral signaling pathways are still not completely understood. For instance, how signaling affects viral transmission across various other cells and tissues remains to be determined. Similarly, how cells fight back to check viral pathogenesis, including sending signals to alert the uninfected cells in the vicinity, is also an exciting area of research. Understanding signaling pathways and their mechanisms to promote virus entry will benefit drug development for virus-specific antiviral treatment.

## 3. Role of Artificial Intelligence in the Study of Virus Infections

Artificial intelligence (AI) is a field of computer science that aims to develop intelligent machines that can perform tasks that typically require human intelligence, such as learning, problem solving, and decision making [91]. AI has the advantage of processing more data in less time, and with ML, the algorithm becomes more efficient as more information is added. In viral pathogenesis research, AI can analyze large datasets, identify new drug targets, and predict viral evolution and drug resistance [92]. In recent years, there have been several advancements in the use of AI in virus–cell interactions, including ML algorithms that can predict viral protein structures [92,93,94,95,96,97,98,99], vaccine design [100], and cross scale microscopy [101]. Deep learning models have been developed that can analyze high-throughput sequencing data, protein–protein interactions, and virus detection [102,103], while natural language processing techniques can analyze scientific literature [104]. In addition, the potential of ML algorithm-based predictions can also be useful when therapies are not working. In this direction, chronic hepatitis C virus (HCV) infection is the leading cause of chronic liver disease and hepatocellular carcinoma. Using multivariable logistic regression (MLR), elastic net (EN), random forest (RF), gradient boosting machine (GBM), and feedforward neural network (FNN) machine learning algorithms, authors reported that direct-acting antivirals (DAA) treatment failure and to determine predictors associated with DAA treatment failure [105]. A detailed ML algorithm workflow of this study in relation to virus–host cell interactions is outlined in Figure 6. Essentially, ML algorithms were also able to effectively predict risk factors, including tobacco, alcohol, and PPIs, and stratification of DAA treatment failure [105].

Big data analysis is an essential component of viral pathogenesis research, as it allows researchers to identify patterns between different variables across multiple test trials quickly. For instance, keeping volume (the amount of data), velocity (the speed at which live data is coming in for analysis), variety (the different data formats), and veracity (the unreliability of the data) as key determinants of big datasets, the strategies are being used to predict disease outbreaks and microbial resistance after combining big data with the public health epidemiology and transmission interfaces to model large-scale ecological drivers, such as regional wildlife biodiversity, human population density, changes in land use, and agricultural industry changes [106]. The latter has resulted in a larger dataset quickly and accurately, allowing researchers to identify new pathogens and potential drug targets [106,107,108]. In addition, several studies have demonstrated the utility of AI in viral pathogenesis research. For instance, AI has been used to predict potential drug targets for the hepatitis C virus (HCV) by analyzing its interaction with human proteins. In this study, the authors showed that their approach could identify previously unknown HCV-host protein interactions and drug targets [105,109]. A different study used ML to identify the genetic factors responsible for several human respiratory infections, including SARS-CoV-2. It was found that their method could accurately predict the host range of coronaviruses, which could help the development of antiviral therapies and future novel vaccine development [110]. Similarly, AI has been used to identify potential drug candidates for treating the Zika virus by analyzing large databases of chemical compounds. This study trained their algorithm on viral sequence data and used it to identify mutations likely to show resistance to existing antiviral drugs. The study highlighted the importance of early surveillance and the need for prediction in the fight against viral infections [111]. Since many viruses infect humans, a ML algorithm using relative synonymous codon usage frequency (RSCU) could detect viral sequences in human metagenomic datasets [112]. The significance of RSCU-based ML techniques can further help identify a large number of putative viral sequences and provide an addition to conventional methods for taxonomic classification (Figure 7) [112].

In the case of SARS-CoV-2, AI has been used to analyze the virus’s genetic sequence, predict the structure of the virus’s spike glycoprotein, and develop early vaccines and therapies [113]. These AI applications have allowed researchers to make faster and more informed decisions in the fight against the COVID-19 pandemic. Staying up to date with the latest research on the usage of AI in the evolution of SARS-CoV-2 is important because it can help healthcare professionals stay informed about the novel variants and their impact on disease progression and or mortality [52]. This can facilitate the discovery of new insights and the development of more effective strategies for combating emerging zoonotic diseases. 

Since more than 70% of emerging infectious diseases are attributed to zoonotic origin, they will continue to be a significant public health concern in the future [114]. With the significant rise in climate change, urbanization, animal migration and trade, vector biology, travel and tourism, anthropogenic factors, and natural factors, we can speculate the emergence, re-emergence, and altered distribution in the global zoonotic footprints affecting new animal species and the human population [115]. In this regard, a recent study analyzed the application of ML and deep learning and the commonly used models and approaches applied to study zoonotic diseases [114]. The study concluded that models such as random forest and eXtreme Gradient Boosting (XGBoost) effectively identified targets despite complex and variable data. Machine learning algorithms such as Random Forest are a combination of tree predictors. Each tree depends on the values of a random vector sampled independently and with the same distribution for all trees in the forest. The generalization error for forests converges as such to a limit as the number of trees in the forest becomes large. The generalization error of a forest of tree classifiers depends on the strength of the individual trees in the forest and the correlation between them [116]. A major benefit of using random forest for prediction modeling is the ability to handle datasets with a large number of predictor variables [117]. On the other hand, the XGBoost-based prediction model utilizes generalizability, low risk of overfitting, and high interpretability; it outperforms other data mining methods for predictive medicine tasks [118]. Additionally, as a virus continues to evolve, it is necessary to have a robust active surveillance program implemented to detect zoonotic spillover events early and accurately so that effective control measures can be taken for public health safety, including one health [119].

Recently, a group developed an AI-powered automated framework for ready detection of the virus-induced cytopathic effect (DVICE) [120]. DVICE uses a CNN (EfficientNet-B0) and transmitted light microscopy images of infected cell cultures, including coronavirus, influenza virus, rhinovirus, HSV, vaccinia virus, and adenovirus [120]. DVICE efficiently measures the virus-induced cytopathic effect (CPE) in different cell types and even in saliva and or serum samples. DVICE provides unbiased infectivity scores of infectious agents causing CPE. This AI based framework can be adapted for laboratory diagnostics, drug screening, serum neutralization or clinical samples [120]. In fact, AI-based platforms can be directly applied to patient diagnosis and early treatment. In this regard, a recent study used a neural network called EBVNet together with H&E-stained slides to aid pathologists in predicting EBV-associated gastric cancer from histopathology [121]. ML is also being used in differential diagnosis to assist clinicians. For instance, a study performed in Korea showed that AI can accurately diagnose esophagitis caused by HSV and or cytomegalovirus [122]. A similar approach has been used in India to diagnose viral keratitis caused by HSV. The latter is significant since many primary care centers do not have laboratory facilities and/or an expert workforce for emergency cases [123]. 

AI has been utilized in the field of ophthalmology to assist in predicting many diseases, such as diabetic retinopathy, age-related macular degeneration, retinopathy of prematurity, glaucoma, and Alzheimer’s disease, through retina imaging analysis [124,125]. Even in the basic virology-based techniques such as electron microscopy and live cell confocal microscopy, the integration of AI-guided computational programming enhances sensitivity toward pathogen detection, multi-imaging datasets, and image restoration capabilities, elucidating previously missing details during host–pathogen interactions [101]. Interestingly, using microscopy-based deep learning methods, AI identified virus infection in a cell without a virus-specific probe. In this study, the authors used adenovirus-infected A549 lung cells and applied a fluorescent bisbenzimide derivative, which binds to double-stranded DNA, as a probe. Live imaging analysis with AI showed a machine-learnable nuclear pattern that could differentiate infected cells ready to disseminate virus (those with lytic nuclei) from non-spreading cells (those with non-lytic nuclei) despite the presence of the same level of GFP-tagged virions. These results provided proof of concept for advancing viral diagnostic methods using deep ML to aid in the study of infection phenotypes such as virus-induced lysis [126].

## 4. General Application of Artificial Intelligence in Infectious Disease

Recent studies with AI have been directed toward the enhancement of clinical applications such as diagnosing diseases via medical imaging techniques, developing new anti-infective drugs, and designing vaccines, enhancing the molecular understanding of transmissible pathogens and combating the emerging threat of global antimicrobial resistance [127,128]. In addition, various ML models including random forest and complex language models have been applied to identify genes and protein–protein interactions associated with host cell changes, predict immunogenicity, and evaluate pathogen killing, host cell adaptation, and virulence [129]. 

Further, ML models have also been used to design synthetic RNAs that mimic messenger RNA, or CRISPR–Cas enzymes for the detection of various types of infectious agents including viruses [129]. For next generation diagnostics convolutional neural networks (CNN) are turning very useful to detect virus infection and in identifying infection-specific features in a clinical sample [126] (Figure 8).

In the infectious disease field, ML models have been used to predict patient outcomes for Ebola virus disease (EVD) [130]. Due to an increase in severe outbreaks of EVD, ML models can estimate mortality risks for their patients using an Ebola Prognostic Score and future symptoms using LR and ANN machine learning models [130,131]. These image analysis and ML techniques are constantly being updated to work with many different mobile devices, which may already contain high-quality cameras, to improve immediate and non-professional diagnoses in the future. The same time-saving approach heavily utilizes ML in drug discovery and targeted therapy development [132]. A recent study predicted cell entry inhibitors of the Ebola virus using ML-based web EBOLApred [93]. The study utilized multiple algorithms and approaches such as random forest (RF), support vector machine (SVM), naïve Bayes (NB), k-nearest neighbor (kNN), and logistic regression (LR). A previous study used similar algorithms to develop hepatitis C virus NS5B and HIV integrase inhibitors [89,133,134]. An HIV study determined that ML had the potential to ultimately accelerate the drug discovery process by whittling down the antiviral compounds that did not work in similar (to HIV) situations [135]. This demonstrates that while converting raw data into ML databases can be time-consuming, the overall benefit of using artificial intelligence far outweighs the time lost in translating data. 

In a separate study, a generative adversarial autoencoder was created to create antiviral drugs for the initial stages of HIV-1 infection. This autoencoder was created from an autoencoder architecture, a virtual compound library of anti-HIV-1 agents, molecular decking of the library’s compounds, molecular fingerprints of the compounds, and a neural network of more than 21 million molecules. This process led to the discovery of new antiviral drugs, which will continue to serve as the basis for future developments in the fight against HIV-1 infections [136]. In both HIV studies, it was proven that AI plays a crucial role in efficiently simplifying antiviral compounds. The general application of AI in virus–host pathogenesis holds tremendous promise for advancing our understanding and response to viral infections. AI’s integrations in clinical settings, vaccine discovery, and surveillance can change how viral diseases are diagnosed and treated. In a recent study, the authors developed AlphaMissense, a deep learning model that predicts the pathogenicity of a missense variant [137]. The success of AI-based prediction will accelerate our understanding of structural studies of highly pathogenic variants for drug discovery and the preparedness for future epidemics and pandemics. Algorithms, prediction models, and image analysis have shown enormous potential in automating the characterization of viral pathogens, saving time and resources while minimizing human error. As AI technologies continue to evolve and become more accessible, they have the potential to transform viral pathology research and contribute to the fight against viral infections with better predictions and high specificity.

## 5. Current Research on Artificial Intelligence and SARS-CoV-2

The COVID-19 pandemic has been one of the most significant public health crises in recent history, causing widespread disruption and an enormous loss of life. However, one positive outcome of the pandemic has been the rapid advancement of research on SARS-CoV-2 and the application of new AI technologies in diagnosis disease prognosis prediction using clinical markers, including predicting antigenic variant evolution [138,139,140]. 

Since the pandemic is officially over, the current state of research is much further along than in the early days of the outbreak. At the start of the pandemic, there was a lot of uncertainty about the virus and its behavior. The initial struggle to understand virus, cell, and tissue-specific receptors aiding in viral spread affecting various cells and tissues resulted in a slow start in developing proper treatments and vaccines. However, as the pandemic continued, researchers were able to collect scientific data and direct evidence about the virus and antigenic behavior on a global scale. The rapid evolution of SARS-CoV-2 research has been aided in part by the urgency of the situation [51]. With the virus causing so much damage and loss of life, there has been a global push to find effective treatments and vaccines as quickly as possible. This sense of urgency has led to new levels of collaboration between researchers worldwide and the sharing of data and resources. The rapid sharing of information has helped researchers make significant strides in understanding the virus. 

Furthermore, the fast pace of research on SARS-CoV-2 has been facilitated by the development of new technologies in the field of AI. AI has shown promise in helping researchers analyze large amounts of data quickly and accurately, allowing them to identify patterns and trends that might not be apparent to the human eye. By using AI to analyze data on SARS-CoV-2, researchers have made significant breakthroughs in understanding the virus and its behavior. One example of this is the identification of SARS-CoV-2 viral entry inhibitors using ML and cell-based pseudotyped particles [90]. They relied on molecular descriptors, which are numerical representations of molecular properties, to build predictive models of a molecule’s activity. Two sets of descriptors were used: atom-type descriptors, which focus on the chemical structure, and 3D fingerprints, which consider the spatial arrangement of atoms. Each type provided complementary information, with atom-type descriptors excelling in predicting compounds similar to known inhibitors and 3D fingerprints performing better for structurally different compounds. Combining these descriptors into a consensus model yielded an improved predictive accuracy (AUC-ROC of 0.91). The area under curve (AUC) measures the ability of the test to correctly classify those with and without the disease, while the Receiver Operator Characteristic (ROC) curve is an excellent way to depict the ability of the test to discriminate disease at each cut point, but in practice, it is also very important to have a single index to summarize the overall performance of the test [141]. Using this model, the researchers screened a library of 173,898 compounds and identified 116 with measurable antiviral activity. AI played an essential role in the study, combining two molecular descriptors into a consensus model and returning the most potent compounds as potential drug candidates.

Another study used baricitinib, an oral Janus kinase inhibitor, previously used for rheumatoid arthritis, to predict its usefulness as a COVID-19 treatment via AI algorithms, which proposed its anti-cytokine effects and as an inhibitor of host cell viral propagation [142]. The predicted biochemical inhibitory effects of baricitinib on human numb-associated kinase were validated, reducing viral infectivity in human primary liver spheroids. A randomized, double-blind, placebo-controlled trial showed that the combination treatment of baricitinib and remdesivir, a previously known effective treatment for hospitalized adult patients with SARS-CoV-2, was superior to remdesivir alone for treating hospitalized COVID-19 patients with pneumonia. These studies led to the combination of baricitinib with remdesivir to produce a more effective therapy against SARS-CoV-2 [143]. The combination treatment resulted in a 1-day shorter recovery time. The benefits of combination treatment were seen across different age, ethnic, and racial groups and were independent of symptom duration or disease severity at enrollment. The faster recovery with baricitinib plus remdesivir suggested that the combination treatment lowered the hospital-associated risk of infections, thrombosis, and errors in hospital drug administration, decreasing the burden on the highly saturated healthcare system. The addition of baricitinib was not associated with thromboembolic events, and patients receiving baricitinib plus remdesivir had a significantly lower incidence of strange events than those who received remdesivir alone. The combination treatment was superior to remdesivir alone in reducing recovery time and accelerating improvement in clinical status, notably among patients who received high-flow oxygen or noninvasive mechanical ventilation. Interestingly, Chan et al., 2021 used ML to identify molecular regulators and therapeutics for targeting SARS-CoV2-induced cytokine release. In this AI-based modeling, authors validated multiple protein kinases, including JAK1, EPHA7, IRAK1, MAPK12, and MAP3K8, as essential downstream mediators of N terminal domains of spike glycoprotein-induced cytokine production, suggesting the significance of multiple signaling pathways in cytokine release [144]. In addition, a recent study utilized bioinformatics analysis and ML to identify and characterize novel viral entry inhibitory peptides against SARS-CoV-2 [145]. Their results were cross-validated via independent testing results, demonstrating the significance of sequence-based features in predicting VEIPs with high accuracy [145].

In a separate study about using AI to predict the interactions of SARS-CoV-2 and human proteins, a Learning Vector Quantization (LVQ) Algorithm was used for feature selection, an essential step to identify characteristics in data [146]. This algorithm identified 1326 human proteins with a 70% probability of potentially interacting with SARS-CoV-2. Many of these interactions already had previous drugs, which shows how AI can reduce data complexity and improve analysis accuracy and efficiency. With the use of AI, the medical community was able to advance its understanding of the SARS-CoV-2 host cell interactions. In a recent study, Elend et al., 2022 used an evolutionary algorithm, an artificial neural network model (EMGA), and molecular dynamics (MD) simulations to design potential drug candidates targeting M^pro^ of SARS-CoV-2 [147]. 

Finally, the widespread infection of SARS-CoV-2 during the pandemic left millions worldwide with long COVID, a heterogeneous condition that remains poorly understood. The latter is described as the constellation of adverse health effects caused by the infection [148]. In long COVID, the virus-generated sequelae affect cardiovascular and neurological areas, including the gastrointestinal tract [149,150,151]. In a recent study, authors identified four reproducible clinical sub-phenotypic patterns of long COVID using deep ML in the selected patient cohort. These sub-phenotypes were characterized by cardiac and renal conditions (sub-phenotype 1), respiratory, sleep, and mood problems (sub-phenotype 2), musculoskeletal and neurological conditions (sub-phenotype 3), and digestive and respiratory system problems (sub-phenotype 4) [35].

## 6. Current Research on Artificial Intelligence in Infectious Disease Population Surveillance to Predict Future Pandemic Scenarios

The public health crisis generated by the novel coronavirus 2019 (COVID-19) has raised an urgent need to develop alternative and emerging technologies to predict and efficiently respond to emergencies such as pandemics and epidemics [152,153]. In this context, the surveillance of infectious disease is a constantly moving field as growing human population needs are impacting the global epidemiological triad, which has brought more vulnerability for unwanted host–pathogen interactions. The new outbreaks of zoonotic diseases are now more visible, threatening overall ecology, including the human population [154]. The role of AI in monitoring and analyzing infectious diseases by identifying and mitigating the key determinants responsible for the outbreaks, tracking disease spread, and developing new prevention strategies are more relevant than ever. 

In this direction, combined development in multiple technologies such as biosensors technology, quantum computing, augmented intelligence, and Generative Pre-trained Transformer 4 have sharply increased our visibility to infectious disease surveillance to prevent its spread [155]. To track emerging virulence in a given pathogen, deep ML focuses on using datasets as how variants of interest affect protein–protein binding. The latter analysis is vital for predicting virus–host interactions, such as the emergence of new pathogenic variants. In this regard, the structural envelope spike protein of SARS-CoV-2 is a classic example that mediates viral entry by interacting with multiple host cell proteins, including sugar molecules [17,57,156,157,158,159,160]. The positive evolutionary selection in spike proteins has led to the new variants having a stronger affinity for one or more proteins, resulting in enhanced severity and mortality in COVID-19 patients. 

In a recently published study, the AI-based framework UniBind was developed, which effectively predicted the binding of spike protein variants to ACE2 receptors and neutralized monoclonal antibodies. In addition, the UniBind platform was also capable of predicting host susceptibility to the emerging variants. Machine learning’s ability to accurately predict protein–protein interactions with immune escape data is undoubtedly relevant since new viruses will continue to evolve. Therefore, such information would be of immense value when developing vaccines and targeted therapy against variants of concern [113]. Because the success of the sustained virus presence in the given population depends on the ability of a viral antigen to escape an immune response, it provides the basis for constant virus evolution. The latter step serves as a key determinant of reinfection rates, disease severity, cross-species spillover, fluctuations in herd immunity, etc. 

To gain further insight, the modular framework EVEscape was developed, which combines fitness predictions from a deep learning model utilizing biophysical and structural information. This platform quantifies the potential of virus escape to the immune response based on the probability of a given mutation maintaining viral fitness, occurring in an antibody epitope, and disrupting antibody binding. The other advantage of EVSscape modeling is that this platform utilizes only information available early in a pandemic, when surveillance sequencing, antibody-antigen structures, or experimental mutational scan data are unavailable. This information is critical for efficient vaccine development. Interestingly, this platform’s effectiveness in identifying escape mutants has been demonstrated against multiple pathogenic viruses such as influenza, Lassa, Nipah, and HIV [161]. In fact, in the face of the current pandemic, other zoonotic diseases caused by Nipah, Langya, monkeypox, Ebola, and influenza-associated outbreaks are also rising in their endemic areas [162,163,164,165,166,167]. The mortality and morbidity risk due to these zoonosis events are far more significant and could be catastrophic to manage the additional crisis due to the shortage of healthcare professionals. In addition, due to the lack of vaccines and therapeutics directed against the abovementioned viral infection except for influenza, strategies to prepare for these potential zoonotic viral outbreaks seem imminent. Such outbreaks would not only affect the human population but also disrupt the affected country’s economic and health infrastructure. Finally, the rapid growth of the human population, climate change, unemployment, and poverty in developing countries have reduced the natural habitat of wild animals and will continue to do so, resulting in their close interaction with humans. This, in turn, will promote the rapid accumulation and adaptability of viral proteins, leading to more spillover events. In this scenario, ML models are highly beneficial and would enable earlier detection of epidemics, including targeted treatment and preventive vaccine therapies to save public health. 

Regarding epidemiological considerations, the core tasks of AI-based health surveillance systems include early prediction of infectious disease outbreaks and biowarfare events, their detection, and public health response modeling and assessment. In this direction, AI-based health surveillance systems will need constant improvement in the accuracy and effectiveness of these systems by refining algorithms and integrating emerging data sources. The emerging threat of the new variants capable of spreading and causing significant mortality remains high [168]. Studies involving deep ML algorithms that can perform genomic mutations, their potential effect on receptor binding and cell spread, and isolation techniques in patient samples are underway. Similarly, the AI-based approach to diagnose and grade patients based on their symptoms has also been widely used among COVID-19 patients [169,170]. 

## 7. Limitations and Challenges of Using Artificial Intelligence in Viral Pathogenesis Research

Although the direct application of AI in medicine is evolving rapidly, the full potential of AI in viral pathogenesis research and its relevance in clinical scenarios remain challenging for multiple reasons. Firstly, the availability of dependable, comprehensive, high- quality datasets plays an essential role in the success of AI models, as limited access to proper datasets can lead to insufficient and unreliable outcomes. Secondly, a lack of adequate training, model validation, and assessment can lead to difficulty in accurately characterizing host–pathogen interactions and their complex mechanisms. For example, infectious diseases involving multiagent pathogens remain a challenge for simulating realistic and dynamic contact networks. Further, a cohesive collaboration among biomedical researchers, data scientists, and clinicians under one domain must be well-coordinated for correctly analyzing and interpreting the molecular data [171,172]. 

Thirdly, the interpretability and explainability of AI models are another challenge. While AI algorithms can generate predictions, the lack of transparency in the decision-making process raises concerns about the reliability and trustworthiness of the results. This can limit research opportunities as it becomes crucial for scientists to understand the underlying reasoning and biological relevance of AI-generated predictions. Without proper interpretability, researchers may misrepresent the distribution affecting the AI-generated data. Lastly, ethical and legal considerations also pose a considerable threat to AI research opportunities. The processes of collection, storage, and analysis of patient data have raised many concerns in the past about privacy, consent, and ownership. Strict adherence to ethical guidelines is necessary to protect patient privacy adequately. These considerations introduce new complexities and limitations that can inhibit the scope of AI research. In addition, the more challenging aspect of an AI-based epidemiological surveillance system is to resolve the ethical and social implications, the model interpretability, and the issues of biases and discrimination in algorithmic decision making to ensure that heterogeneous populations are fairly covered. The public trust and widespread adoption of AI technologies in public health and clinical medicine offer a significant advantage in early disease prevention and control [102].

## 8. Conclusions

AI enables researchers to uncover hidden insights and generate models that aid in identifying mechanistic details of virus–host cell interactions to develop novel and efficacious vaccine and or treatments against infectious agents [100,110,111,173]. By combining ML and its ability to process substantial amounts of data originating from computational biology, AI has been shown to speed up the discovery and development of novel drugs. AI analysis also has the potential to lead to obscure discoveries and an overall more profound understanding of viral pathogenesis and future emerging threats (Table 1). At the same time, it is clear that emerging new viruses and their host cell interactions will require meticulous attention to develop innovative approaches to understand the topography of virus membrane and their ability to interact with one or multiple cellular receptors and nearby molecules. If we can generate a viral entry map for all the pathogenic viruses, including the associated signaling pathways, we can develop novel interventions with the help of AI-based technologies to prevent infections. In this regard, dissecting viral proteins and host cell receptors by conducting structure–function analysis would further contribute to understanding virus pathogenesis and the associated cellular tropism. A shared epitope across multiple viruses would encourage the development of a broad-spectrum antiviral [44]. Understanding virus endosomal trafficking would translate to developing a post-entry inhibitor, which would protect uninfected cells from being infected.

Current research in viral entry has witnessed significant advancements with the integration of AI. The role of AI in this field has been instrumental in creating novel approaches and tools that can help improve traditional methods and enhance the current understanding of viral pathogenesis and efficient diagnostics (Table 1). The advantages of using AI in virus–host interactions are clear: AI algorithms possess the capability to quickly and efficiently process large quantities of data, and these algorithms can recognize patterns that may not be obvious to humans, and AI allows for the cross-usage of many different diverse data sources to create a fully comprehensive approach towards studying viral infections. This advances the research process and allows for rapid analysis and, at times, new interpretations. On the other hand, there are some challenges in AI-based technologies that persist in biomedical research. Examples include collecting reliable data, developing unbiased and reliable algorithms, copyright issues, and implementing translational research that could benefit healthcare settings. 

Integrating AI with other surveillance methods, such as traditional laboratory testing and epidemiological investigations, can also enhance global viral surveillance capabilities. AI’s numerous abilities in data analysis and prediction services will make it much easier for researchers to comprehensively understand viral patterns to improve the overall understanding of viral pathogenesis.

However, it is necessary to address the limitations that are associated with the use of AI in viral pathogenesis research: ensuring data accessibility and quality is essential for reliable results; the lack of transparency in AI’s “thinking process” can lead to a lack of accountability if there are problems; many AI models cannot keep up with rapidly evolving strains and therefore cannot work for the most modern pathogens; and, lastly, the ethical and legal considerations must be addressed to ensure trust. By actively solving these challenges and implementing strategies to improve data accessibility, enhance model transparency, adapt to evolving pathogens, and follow ethical guidelines, researchers can fully harness the potential of AI in pathogenesis research. The responsible integration of AI in viral host pathogenesis is incredibly promising. From its numerous capabilities, researchers can accelerate the understanding of viral pathogens and hasten the discovery of effective preventative measures against infections. The collaborative partnership between human expertise and the strengths and weaknesses of technology will pave the way for a brighter future in the fight against viral pathogens.

## Figures and Tables

**Figure 1 biomolecules-14-00911-f001:**
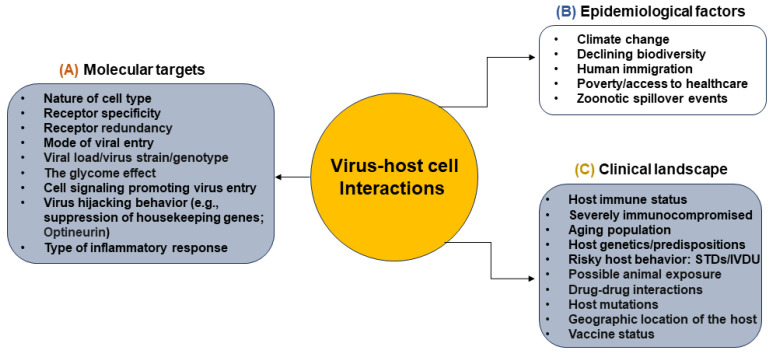
The multi-factorial dependency and associated complexity of virus–host cell interactions. The schematic diagram highlights the significance of the disease at the molecular level (panel (**A**)) affected by epidemiological factors (panel (**B**)) along with host behavior/immune status, etc. (panel (**C**)), impacting the overall patient outcomes during virus–host cell interactions.

**Figure 2 biomolecules-14-00911-f002:**
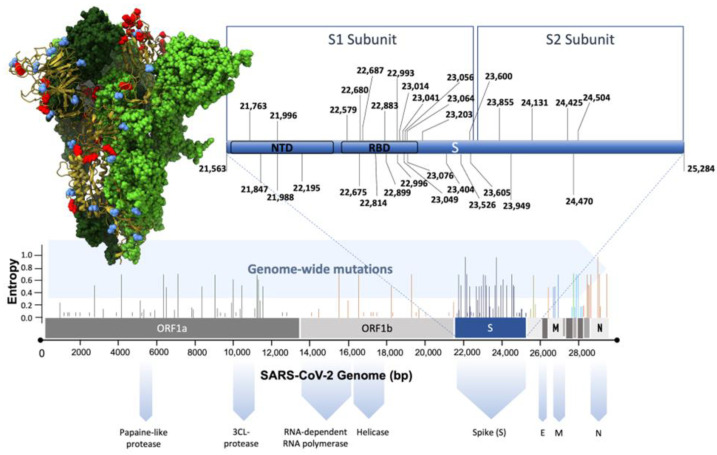
Genomic mutations in SARS-CoV-2-impacting domains of the spike glycoprotein are highlighted. Shannon entropy plots identify the sites with the highest mutagenic potential (taken from the Nextstrain database). An amplification of the S gene is shown in blue; this gene is the part of the genome with the highest number of mutations. Within this gene, the highest mutation prevalence is in the S1 subunit, specifically in the receptor binding domain (RBD). A 3D view of the spike protein shows the mutations in red and the glycosylation sites in blue. The image was generated using Visual Molecular Dynamics 704 (VMD).

**Figure 3 biomolecules-14-00911-f003:**
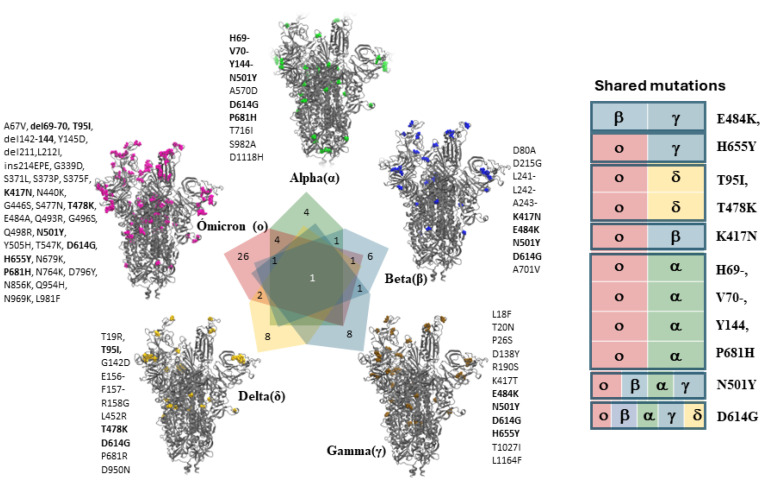
Unique shifts in the amino acid residues present on the S1 and S2 domains in the spike glycoproteins show a higher frequency of SARS-CoV-2 mutations. The coded colors indicate the mutations present in the highlighted variants. The table in the right panel shows the mutations shared between the variants. The images were generated using Visual Molecular Dynamics 704 (VMD).

**Figure 4 biomolecules-14-00911-f004:**
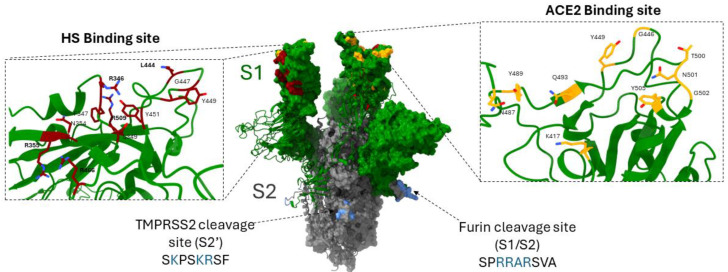
A critical determinant of SARS-CoV-2 tropism and infectivity is the spike glycoprotein with two subdomains. The S1 domain (shown in green) and S2 domain (shown in gray) interactions with the host cell receptors are highlighted. In the left panel, the red regions in the S1 domain interact with cell surface heparan sulfate (HS), while the right panels show that the yellow residues in the S1 domain interact with the ACE-2 receptor. The highlighted bottom blue residues in S2 domains represent the cleavage sites of the proteases TMPRSS2 and Furin. The images were generated using Visual Molecular Dynamics 704 (VMD).

**Figure 5 biomolecules-14-00911-f005:**
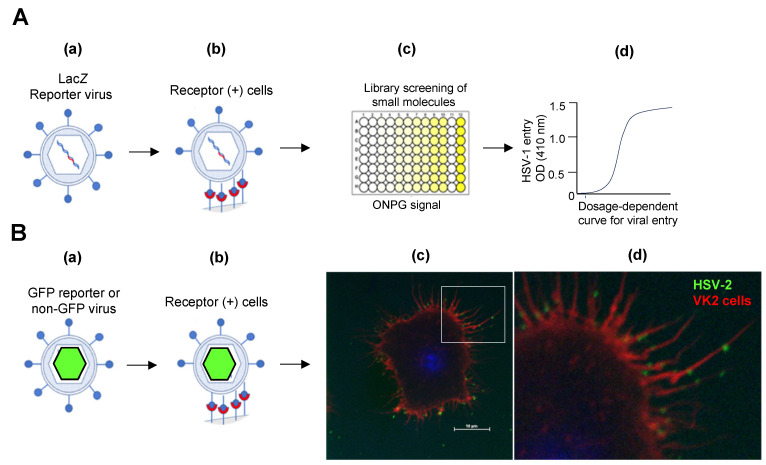
Reporter-based viruses provide a quantitative platform to understand virus–host interactions at the molecular level, including rapid high-throughput screening for potential antiviral compounds. (**A**) Schematic representation of a lac*Z* (β-galactosidase) encoded reporter herpes simplex virus type-1 (HSV-1). Upon entry into receptor-positive cells (sub-panels (**a**,**b**)) results in a quantifiable ONPG signals (yellow-colored multi-wells) using a plate reader (sub-panels (**c**,**d**)). The receptor-negative cells infected with the virus show no signals for ONPG assay (colorless multi-well; sub-panels (**c**,**d**)) (**B**) Imaging platform using either capsid GFP-tagged virus or non-GFP tagged virus (sub-panels (**a**,**b**)) provide a valuable tool to assess virus localization under live or fixed cell imaging (sub-panels (**c**,**d**)). This representative fixed image was captured using vaginal epithelial cells (VK2) infected with HSV-2 using FITC labeled anti-gD antibody and red phalloidin staining. The initial stages of virus localization on active filopodia suggest viral surfing leads to faster virus distribution on the host cell (sub-panels (**c**,**d**)). Previous studies have demonstrated the HSV surfing mechanism on filopodia [32,84].

**Figure 6 biomolecules-14-00911-f006:**
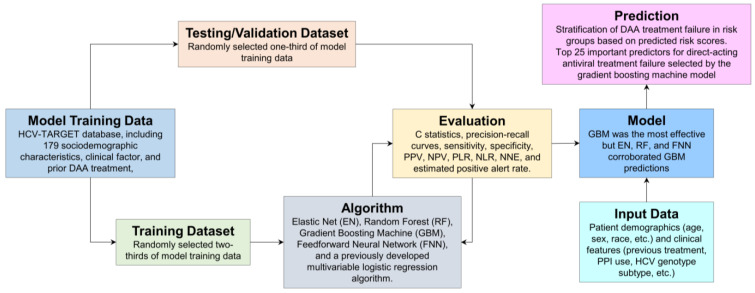
An overview of the machine learning (ML) algorithm for predicting direct-acting antivirals (DAAs) treatment failure in chronic hepatitis C infection. Data preparation involved gathering patient data from the HCV-TARGET registry, including demographics, clinical features, and treatment history. A model was developed using prior ML models (Elastic Net, Random Forest, Gradient Boosting Machine, Feedforward Neural Network) to handle complex interactions and numerous predictors associated with DAA treatment outcomes. The evaluation used a validation dataset to assess the models’ predictive performance, focusing on metrics such as C-statistics, sensitivity, specificity, and other predictive values. The algorithm iteratively self-refined the models based on evaluation results to enhance predictive accuracy. The finished model can stratify DAA treatment failure based on risk scores calculated from 25 potential predictors. This exemplifies the stages of applying ML to classify virus–host interactions [105].

**Figure 7 biomolecules-14-00911-f007:**
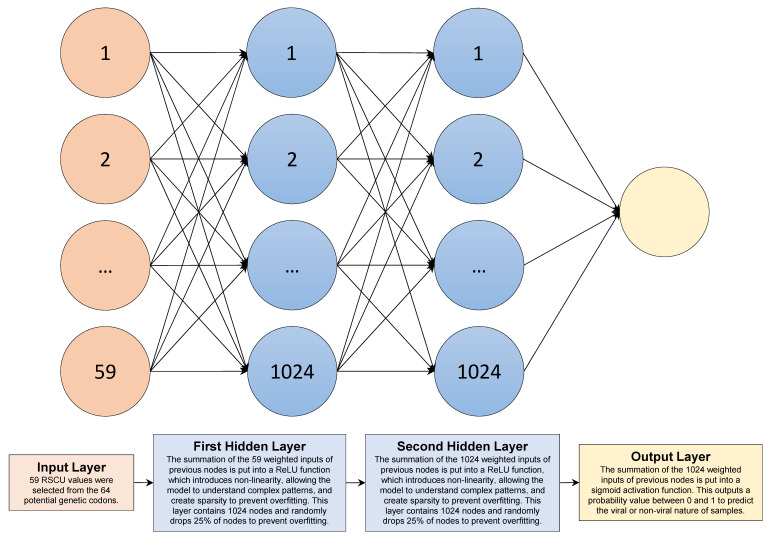
Illustration of a feedforward neural network for detecting viral sequences in human metagenomic datasets. Input data based on 59 calculated relative synonymous codon usage (RCSU) calculated values. Using two fully connected hidden layers, each containing 1024 nodes, the algorithm simplified data using ReLU nonlinearity, a 25% random node dropout rate, and class weighting. The final output layer uses a sigmoid activation function to produce a probability value between 0 and 1, indicating the likelihood of a sequence being of viral origin [112].

**Figure 8 biomolecules-14-00911-f008:**
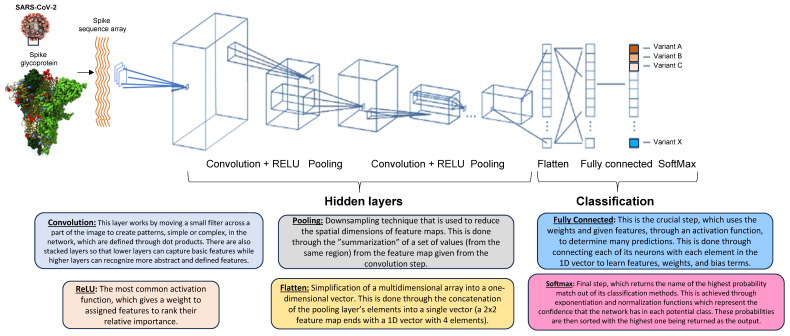
An overview of a Convolutional Neural Network (CNN). This figure provides a visual representation of a CNN workflow. Throughout the hidden layers, features are calculated through convolution and pooling layers; eventually, each is given a specified weight for importance. In the classification layers, these features are simplified into a one-dimensional vector and used to calculate predictions; lastly, the network’s output is converted into class probabilities where the highest probability’s class name is returned. For virus–host interaction systems, CNNs can be used to predict outbreaks of potentially dangerous viral variants (e.g., SARS-CoV-2) by efficient image classification with thousands of collected databases in a standardized manner.

**Table 1 biomolecules-14-00911-t001:** Current significance of artificial intelligence (AI) in the study of virus–host cell interactions.

AI Applications in the Field of Virus-Host Cell Interactions	References
Understanding and predicting key components of viral proteins facilitating cell entry	[52,92,93,94,95,96,97,98,99]
Predictions about the commonality and diversity among the viral entry receptors	[44]
Glycan immunogenicity and pathogenicity, and glycan-mediated immune evasion	[27,28,29]
Glycan motifs involved in virus-host cell interactions	[20,27,28,29,43]
Understanding viral tropism, pathogenesis, and cross-species transmissibility	[44,61,62,63,114,115,168]
Designing of novel antiviral drugs targeting viral entry and vaccine design	[100,110,145]
Identifying molecular regulators and therapeutics for targeting virus induced cytokine release	[144]
Laboratory diagnostics, drug screening, serum neutralization	[120]
Predictions for treatment failure with the antiviral drugs	[105]

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
