# Peer review of "Significance of Artificial Intelligence in the Study of Virus–Host Cell Interactions"

_biomolecules, 2024, doi:10.3390/biom14080911_

Round 1

Reviewer 1 Report

Comments and Suggestions for Authors

Comments on the Quality of English Language

Author Response

Dear Reviewer-1

We would like to thank you for a prompt review of our manuscript. We are pleased to inform you that we have made the required revisions to the original manuscript by incorporating the required changes.  All the changes made to the revised manuscript are highlighted in red. Below is the itemized description of changes made in the revised manuscript as per the reviewer’s suggestions.

Reviewer 1

  1. Where’s the AI? The title creates an expectation that the work is about AI, but AI does not get mentioned until 40 percent into the paper in section 3.

Response: As suggested by the reviewer we have brought up the AI’s based modeling work in relation to virus-host cell interactions throughout the manuscript from section 1 to section 8. All the changes made to the revised manuscript are highlighted in red.

  1. Where are the virus-host interactions? The first two sections of the paper focus on mechanisms of virus entry, only a subset of ways viruses interact with their host cells. After AI is introduced in section 3 and beyond, many other facets of virus-host interaction and disease enter the picture. It is unclear why there is such a focus on virus entry for the first two sections while the AI examples are much broader.

Response: As stated above we have balanced the review contents by bringing  necessary AI sections throughout the manuscript. Additionally, we have added a table to highlight the current significance of AI in virus pathogenesis.  Again, all the changes made to the revised manuscript are highlighted in red.

  1. Page 5, line 155. The opening sentence of the paragraph creates the expectation that we will learn about roles of cell-to-cell contacts in virus entry: “Some viruses enter the cells via direct cell-to-cell contacts at the cell junction where polarized cytoskeleton, cell adhesion molecules, and viral proteins use virological synapses to access the host cell [1,15,37].” However, this expectation is not met. The remainder of the paragraph describes virus-cell interactions and their roles in entry.

Response: We have edited the paragraph to connect better with the rest of the paragraphs.

  1. The abstract could better reflect the title. Artificial intelligence (AI) is the focus of the title, but AI and ML are only mentioned at the end of the abstract. Readers expect the abstract to tell us how AI or ML are contributing to new treatments and prevention strategies.

Response: We agree with the reviewer’s comment. In this direction we have focused on highlighting AI, ML and DL in our abstract including adding new and relevant information throughout the text. All the changes made throughout the manuscript are highlighted in red.

  1. Page 2, 3: It is stated that negatively charged heparan sulfate proteoglycans (HSPGs) are broadly used by microbes and viruses to gain entry to host cells, so targeting HSPGs may offer a new kind of broad acting antimicrobial treatment. However, there is no discussion of potential downsides to such strategies. For example, HSPGs are ubiquitous on the surface of many cell types and play a role in various physiological processes, such as cell adhesion, growth, and repair. Targeting HSPGs could inadvertently affect these essential functions, leading to unintended side effects.

Response:  We agree with the reviewer’s comment, hence have restated our previous comments on targeting HSPGs. In addition, we also have expanded the usage of AI in glycans in predicting virulence and immunity etc.

  1. Lines 206-219. Signaling pathways and receptor-mediated virus internalization. This paragraph suggests that inducing an oncogenic response is beneficial for viruses and that such processes are linked to receptor-mediated virus internalization. But many viruses employ receptor-mediated virus internalization without being oncogenic. Further, the subsequent details on HSV-2 imply it is oncogenic; however, the evidence for oncogenic properties of dsDNA viruses is better established for Human Papillomavirus, Hepatitis B virus, and Epstein-Barr Virus. So, the paragraph is misleading in focusing on HSV-2.

Response: As per reviewer suggestions, we have added appropriate examples of HPV and Hep B viruses with references. The HSV-2 example in context to HPV and Hep B is relevant since many unrelated viruses use actin mediated signaling to favor cell entry. In this direction we also have provided the relevant references.

  1. Lines 275-279. Please cite references to support the underlined claim as indicated: In recent years, there have been several advancements in the use of AI in virus-cell interactions, including machine learning algorithms that can predict viral protein structures (cite references), deep learning models that can analyze high-throughput sequencing data (cite references) (Fig.6), and natural language processing techniques that can analyze scientific literature (cite references). Further, the mention of Figure 6 here suggests the figure will illustrate how AI can be applied to virus data; however, the actual figure is “An overview of the Machine Learning Algorithm Workflow,” with no mention of virology.

Response: We have added all the missing references in the text including updating Fig. 6 which focuses on Hepatitis C virus. Also, we have added the reference for Fig. 6.

  1. Lines 291-307. Please provide more specifics on what AI methods were used, how the methods aligned with the needs of the studies, and to what extent the predictions were validated with experimental data.

Response: We have now provided more specifics with AI methods including their predictions with the experimental data.

  1. Line 293-294. “AI can be used to analyze large datasets quickly and accurately, allowing researchers to identify new pathogens and potential drug targets (Fig. 7).” This text creates an expectation that we will see in Fig. 7 how AI can help identify new pathogens and potential drug targets; however, Fig. 7 has the caption “Illustration of a feedforward neural network architecture” and it has no mention of new pathogens or drug targets.

Response: We have corrected the statement and also the caption of Fig. 7.

  1. Lines 329-330: “The study concluded that models such as random forest and eXtreme Gradient Boosting (XGBoost) effectively identified targets despite complex and variable data.” Explain for your readers how these AI models work, why they are appropriate for the desired research goals, and the extent to which they have been validated by experiments.

Response: We have explained the significance of random forest and eXtreme Gradient Boosting (XGBoost) models for achieving the desired goals and validation.

  1. Lines 335-347: Are references 67-to-70 examples of natural language processing that was mentioned earlier (point 6 above)? If so, help the reader connect the dots.

Response: As suggested by the reviewer, we have tried connecting the statements.

  1.  Lines 367-379 refer to malaria, microscope blood smears, an application of CNN, and Fig. 7. However, Fig. 7 shows SARS-CoV-2 (not malaria), and the caption refs say nothing about blood smears.

Response: As suggested by the reviewer, we have corrected the lines. Fig. 7 has also been updated in reference to SARS-CoV-2 with appropriate reference.

  1.  Line 374: “CNNs use convolutional layers to learn and extract features from images while working as highly efficient, detailed image recognition software.” The “C” in CNNs stands for “convolutional,” so avoid using the term in explaining how CNNs work.

Response: As suggested by the reviewer, we have deleted the portion describing how CNNs work.

  1. Line 463-464. “Combining these descriptors into a consensus model yielded an improved predictive accuracy (AUC-ROC of 0.91).” The authors never define the concept of the AUC or ROC, which one would expect for a thoughtful review on AI or ML methods.

Response: We have now defined the significance of AUC and ROC with appropriate references.

  1.  Section 7 (Lines 579-607). This section on the limitations and challenges of AI has the potential to be very useful to virologists, clinicians, and others. However, no references are provided.

 Response: We have now provided the references for section 7.

  1.  Complete sections on Funding (Line 668), Data (672), COI (680).

Response: We have now completed the sections on funding, data and COI.

  1. Figures 2-through-5 appear to be adapted from the literature, but the figure captions cite no sources.

Response: Fig. 2- Fig. 5A have not been published or adapted from the literature, figure captions already state that the data were obtained from Nexstrain database, and the figures obtained by VMD. For Fig. B (panels c and d) we have provided the reference from previously published data.

Reviewer 2 Report

Comments and Suggestions for Authors

The manuscript by Akash Saini with co-authors describes the role of artificial intelligence (AI) in solving problems related to virus-host interactions. Despite the significant contribution to this area of research, the manuscript has some major drawbacks.

1) The Title and Abstract suggest that the review focused on interactions of viruses with host cells, especially viral entry; however, the text contains description of AI application to other tasks such as epidemic surveillance (section 6). Moreover, the text contains examples related to bacterial and protozoal infections, e.g., CNN based image analysis for malaria diagnosis. I suggest the authors to either shift the focus of review to virus-host interactions in general (including viral escape from immune response, viral evolution, understanding of host dependence and restriction factors, etc), or include more examples of applying AI to data on viral entry into host cells and viral-induced cell signaling.

2) The text in the sections “3. Role of artificial intelligence in virus infections” and “4. General application of artificial intelligence in infectious disease” is poorly structured. It is unclear the differences between two sections. Both sections contain examples related to viral infections. The figure 7 has no relation to the text and can be removed. Instead, it would be better to provide a table with summary of AI application in virus-host interactions: the tasks solved by AI (search for viral enzymes inhibitors, prediction of virus evolution, diagnostics based on images, prediction of interaction between viral and host proteins, etc), the AI and machine learning methods applied, the training sets used, references to original publications. Generally, more examples regarding virus-cell interactions, especially viral entry, should be provided. Particularly, an example on CNN application (page 11) should be related to viral infections instead of malaria.

3) The review contains very few examples of AI application in the field of viral-host interactions. For instance, there are published many studies related to prediction of interactions between SARS-CoV-2 proteins and human proteins as well as studies related to the search of coronavirus inhibitors. Indeed, it is impossible to describe all of them in details in the text, however, brief description with references should be provided. In contrary, the current examples in sections 3-6 should contain more details regarding training data and AI methods, since the major bottleneck in the application of AI is an absence or low quality of data to build models. For example, the authors describe the study related to the search for SARS-CoV-2 entry inhibitors by QSAR – based method (page 13, lines 452-467). At the beginning of the pandemic, there was no known inhibitors. Currently, the existing data on inhibitors in public databases such as ChEMBL and PubChem is controversial, so a compound may by “active” in some studies and “inactive” in others. This prevents the effective use of AI to find inhibitors of SARS-CoV-2 life cycle.

4) The Figure 5 contains description of experimental results. Have they been previously published? If yes, please, provide corresponding reference. If no, it would be better to remove them from the review, as new unpublished data is generally not suitable for reviews.

Author Response

Dear Reviewer-2

We would like to thank you for a prompt review of our manuscript. We are pleased to inform you that we have made the required revisions to the original manuscript by incorporating the required changes.  All the changes made to the revised manuscript are highlighted in red. Below is the itemized description of changes made in the revised manuscript as per the reviewer’s suggestions.

Reviewer 2

  1. The Title and Abstract suggest that the review focused on interactions of viruses with host cells, especially viral entry; however, the text contains description of AI application to other tasks such as epidemic surveillance (section 6). Moreover, the text contains examples related to bacterial and protozoal infections, e.g., CNN based image analysis for malaria diagnosis. I suggest the authors to either shift the focus of review to virus-host interactions in general (including viral escape from immune response, viral evolution, understanding of host dependence and restriction factors, etc), or include more examples of applying AI to data on viral entry into host cells and viral-induced cell signaling.

Response: As suggested by the reviewer we have deleted examples related to protozoans or bacteria in the revised manuscript. The updated review mostly focuses on viral entry and cell signaling with relevant references. 

  1. The text in the sections “3. Role of artificial intelligence in virus infections” and “4. General application of artificial intelligence in infectious disease” is poorly structured. It is unclear the differences between two sections. Both sections contain examples related to viral infections. Figure 7 has no relation to the text and can be removed. Instead, it would be better to provide a table with summary of AI application in virus-host interactions: the tasks solved by AI (search for viral enzymes inhibitors, prediction of virus evolution, diagnostics based on images, prediction of interaction between viral and host proteins, etc), the AI and machine learning methods applied, the training sets used, references to original publications. Generally, more examples regarding virus-cell interactions, especially viral entry, should be provided. Particularly, an example on CNN application (page 11) should be related to viral infections instead of malaria.

Response: As suggested by the reviewer we have restructured the general application of artificial intelligence in infectious diseases. We have also improved both Fig. 6 and Fig. 7 connecting with the text and also providing specific examples to virus. In addition, we also have provided Table 1 which highlights the summary of AI’s application during virus-host interactions along with the references. As suggested, we have used a virus example for CNN application. 

  1. The review contains very few examples of AI application in the field of viral-host interactions. For instance, there are published many studies related to prediction of interactions between SARS-CoV-2 proteins and human proteins as well as studies related to the search of coronavirus inhibitors. Indeed, it is impossible to describe all of them in detail in the text, however, brief description with references should be provided. In contrary, the current examples in sections 3-6 should contain more details regarding training data and AI methods, since the major bottleneck in the application of AI is an absence or low quality of data to build models. For example, the authors describe the study related to the search for SARS-CoV-2 entry inhibitors by QSAR – based method (page 13, lines 452-467). At the beginning of the pandemic, there were no known inhibitors. Currently, the existing data on inhibitors in public databases such as ChEMBL and PubChem is controversial, so a compound may by “active” in some studies and “inactive” in others. This prevents the effective use of AI to find inhibitors of SARS-CoV-2 life cycle.

Response: The updated version of the revised article contains additional and relevant examples of AI application in general with multiple other viruses including SARS-CoV-2.

  1. The Figure 5 contains description of experimental results. Have they been previously published? If yes, please, provide corresponding reference. If no, it would be better to remove them from the review, as new unpublished data is generally not suitable for reviews.

Response: Fig. 5A and Fig. B (panels a and b) have not been published or adapted from the literature. For Fig. B (panels c and d) we have provided the reference from previous published findings in the figure legend.

Round 2

Reviewer 1 Report

Comments and Suggestions for Authors

The authors have adequately addressed prior suggestions and concerns.